# Generalized Hindsight for Reinforcement Learning

**Alexander C. Li**
University of California, Berkeley
alexli1@berkeley.edu

**Lerrel Pinto**
New York University
lerrel@cs.nyu.edu

**Pieter Abbeel**
University of California, Berkeley
pabbeel@cs.berkeley.edu

## Abstract

One of the key reasons for the high sample complexity in reinforcement learning (RL) is the inability to transfer knowledge from one task to another. In standard multi-task RL settings, low-reward data collected while trying to solve one task provides little to no signal for solving that particular task and is hence effectively wasted. However, we argue that this data, which is uninformative for one task, is likely a rich source of information for other tasks. To leverage this insight and efficiently reuse data, we present Generalized Hindsight: an approximate inverse reinforcement learning technique for relabeling behaviors with the right tasks. Intuitively, given a behavior generated under one task, Generalized Hindsight returns a different task that the behavior is better suited for. Then, the behavior is relabeled with this new task before being used by an off-policy RL optimizer. Compared to standard relabeling techniques, Generalized Hindsight provides a substantially more efficient re-use of samples, which we empirically demonstrate on a suite of multi-task navigation and manipulation tasks. (Website[1])

## 1 Introduction

Model-free reinforcement learning (RL) combined with powerful function approximators has achieved remarkable success in games like Atari [43] and Go [64], and control tasks like walking [24] and flying [33]. However, a key limitation to these methods is their sample complexity. They often require millions of samples to learn simple locomotion skills, and sometimes even billions of samples to learn more complex game strategies. Creating general purpose agents will necessitate learning multiple such skills or strategies, which further exacerbates the inefficiency of these algorithms. On the other hand, humans (or biological agents) are not only able to learn a multitude of different skills, but from orders of magnitude fewer samples [32]. So, how do we endow RL agents with this ability to learn efficiently across multiple tasks?

One key hallmark of biological learning is the ability to learn from mistakes. In RL, mistakes made while solving a task are only used to guide the learning of that particular task. But data seen while making these mistakes often contain a lot more information. In fact, extracting and re-using this information lies at the heart of most efficient RL algorithms. Model-based RL re-uses this information to learn a dynamics model of the environment. However for several domains, learning a robust model is often more difficult than directly learning the policy [15], and addressing this challenge continues to remain an active area of research [46]. Another way to re-use low-reward data is off-policy RL, where in contrast to on-policy RL, data collected from an older policy is re-used while optimizing the new policy. But in the context of multi-task learning, this is still inefficient since data generated

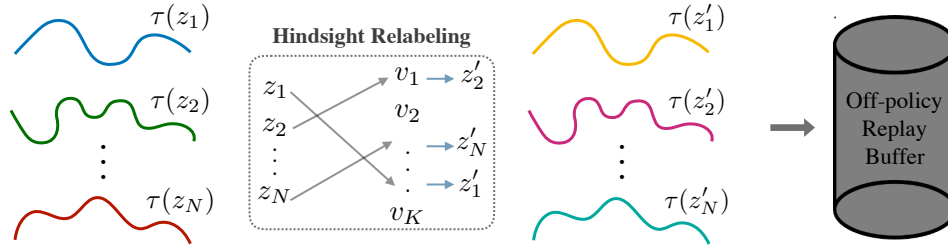

Figure 1: Trajectories $\tau(z_i)$, collected trying to maximize $r(\cdot|z_i)$, may contain very little reward signal about how to solve their original tasks. Generalized Hindsight checks against randomly sampled "candidate tasks" $\{v_i\}_{i=1}^K$ to find different tasks $z_i'$ for which these trajectories can serve as "pseudo-demonstrations." Using off-policy RL, we can obtain stronger reward signal from these relabeled trajectories.

from one task cannot effectively inform a different task. Towards solving this problem, work by Andrychowicz et al. [2] focuses on extracting even more information through *hindsight*.

In goal-conditioned settings, where tasks are defined by a sparse goal, Hindsight Experience Replay (HER) [2] relabels the desired goal, for which a trajectory was generated, to a state seen in that trajectory. Therefore, if the goal-conditioned policy erroneously reaches an incorrect goal instead of the desired goal, we can re-use this data to teach it how to reach this incorrect goal. Hence, a low-reward trajectory under one desired goal is converted to a high-reward trajectory for the unintended goal. This new relabeling provides a strong supervision and produces significantly faster learning. However, a key assumption made in this framework is that goals are a sparse set of states that need to be reached. This allows for efficient relabeling by simply setting the relabeled goals to the states visited by the policy. But for several real world problems like energy-efficient transport, or robotic trajectory tracking, rewards are often complex combinations of desirables rather than sparse objectives. So how do we use hindsight for general families of reward functions?

In this paper, we build on the ideas of goal-conditioned hindsight and propose **Generalized Hindsight**. Here, instead of performing hindsight on a task-family of sparse goals, we perform hindsight on a task-family of reward functions. Since dense reward functions can capture a richer task specification, GH allows for better re-utilization of data. Note that this is done along with solving the task distribution induced by the family of reward functions. However, for relabeling, instead of simply setting visited states as goals, we now need to compute the reward functions that best explain the generated data. To do this, we draw connections from Inverse Reinforcement Learning (IRL), and propose an *Approximate IRL Relabeling* algorithm we call AIR. Concretely, AIR takes a new trajectory and compares it to $K$ randomly sampled tasks from our distribution. It selects the task for which the trajectory is a "pseudo-demonstration," i.e. the trajectory achieves higher performance on that task than any of our previous trajectories. This "pseudo-demonstration" can then be used to quickly learn how to perform that new task. We illustrate the process in Figure 1. We test our algorithm on several multi-task control tasks, and find that AIR consistently achieves higher asymptotic performance using as few as 20% of the environment interactions as our baselines. We also introduce a computationally more efficient version, which relabels by comparing trajectory rewards to a learned baseline, that also achieves higher asymptotic performance than our baselines.

In summary, we present three key contributions in this paper: (a) we extend the ideas of hindsight to the generalized reward family setting; (b) we propose AIR, a relabeling algorithm using insights from IRL. This connection has been concurrently and independently studied in [17], with additional discussion in Section 4.5; (c) we demonstrate significant improvements in multi-task RL on a suite of multi-task navigation and manipulation tasks.

## 2 Background

Before discussing our method, we briefly introduce some background for multi-task RL and Inverse Reinforcement Learning (IRL). For brevity, we defer basic formalisms in RL to Appendix A.

### 2.1 Multi-Task RL

A Markov Decision Process (MDP) $\mathcal{M}$ can be represented as the tuple $\mathcal{M} \equiv (\mathcal{S}, \mathcal{A}, \mathcal{P}, r, \gamma, \mathbb{S})$, where $\mathcal{S}$ is the set of states, $\mathcal{A}$ is the set of actions, $\mathcal{P} : \mathcal{S} \times \mathcal{A} \times \mathcal{S} \to \mathbb{R}$ is the transition probability

function, $r : \mathcal{S} \times \mathcal{A} \to \mathbb{R}$ is the reward function, $\gamma$ is the discount factor, and $\mathbb{S}$ is the initial state distribution.

The goal in multi-task RL is to not just solve a single MDP $\mathcal{M}$, but to solve a distribution of MDPs $\mathcal{M}(z)$, where $z$ is the task-specification drawn from the task distribution $z \sim \mathcal{T}$. Although $z$ can parameterize different aspects of the MDP, we are specially interested in different reward functions. Hence, our distribution of MDPs is now $\mathcal{M}(z) \equiv (\mathcal{S}, \mathcal{A}, \mathcal{P}, r(\cdot|z), \gamma, \mathbb{S})$. Thus, a different $z$ implies a different reward function under the same dynamics $\mathcal{P}$ and start distribution $\mathbb{S}$. One may view this representation as a generalization of the goal-conditioned RL setting [61], where the reward family is restricted to $r(s, a|z = g) = -d(s, z = g)$. Here $d$ represents the distance between the current state $s$ and the desired goal $g$. In sparse goal-conditioned RL, where hindsight has previously been applied [2], the reward family is further restricted to $r(s, a|z = g) = \mathbb{1}[d(s, z = g) < \epsilon]$. Here the agent gets a positive reward only when $s$ is within $\epsilon$ of the desired goal $g$.

## 2.2 Hindsight Experience Replay (HER)

HER [2] is a simple method of manipulating the replay buffer used in off-policy RL algorithms that allows it to learn state-reaching policies more efficiently with sparse rewards. After experiencing some episode $s_0, s_1, ..., s_T$, every transition $s_t \to s_{t+1}$ along with the goal for this episode is usually stored in the replay buffer. However with HER, the experienced transitions are also stored in the replay buffer with different goals. These additional goals are states that were achieved later in the episode. Since the goal being pursued does not influence the environment dynamics, one can replay each trajectory using arbitrary goals, assuming we optimize with an off-policy RL algorithm [57].

## 2.3 Inverse Reinforcement Learning (IRL)

In IRL [48], given an expert policy $\pi_E$ or, more practically, access to demonstrations $\tau_E$ from $\pi_E$, we want to recover the underlying reward function $r^*$ that best explains the expert behaviour. Although there are several methods that tackle this problem [58, 1, 72], the basic principle is to find $r^*$ such that:

$$\mathbb{E}[\sum_{t=0}^{T-1} \gamma^t r^*(s_t)|\pi_E] \geq \mathbb{E}[\sum_{t=0}^{T-1} \gamma^t r^*(s_t)|\pi] \;\; \forall \pi \tag{1}$$

We use this framework to guide our *Approximate IRL* relabeling strategy for Generalized Hindsight.

# 3 Generalized Hindsight

## 3.1 Overview

Given a multi-task RL setup, i.e. a distribution of reward functions $r(.|z)$, our goal is to maximize the expected reward $\mathbb{E}_{z \sim \mathcal{T}}[R(\pi|z)]$ across the task distribution $z \sim \mathcal{T}$ through optimizing our policy $\pi$. Here, $R(\pi|z) = \sum_{t=0}^{T-1} \gamma^t r(s_t, a_t \sim \pi(s_t|z)|z)$ represents the cumulative discounted reward under the reward parameterization $z$ and the conditional policy $\pi(.|z)$. One approach to solving this problem would be the straightforward application of RL to train the $z-$ conditional policy using the rewards from $r(.|z)$. However, this fails to re-use the data collected under one task parameter $z$ $(s_t, a_t) \sim \pi(.|z)$ to a different parameter $z'$. In order to better use and share this data, we propose to use hindsight relabeling, which is detailed in Algorithm 1.

---

**Algorithm 1** `Generalized Hindsight`

---

1: **Input:** Off-policy RL algorithm $\mathbb{A}$, strategy $\mathbb{S}$ for choosing suitable task variables to relabel with, reward function $r : \mathcal{S} \times \mathcal{A} \times \mathcal{T} \to \mathbb{R}$
2: **for** episode = 1 to $M$ **do**
3:      Sample a task variable $z$ and an initial state $s_0$
4:      Roll out policy on $z$, yielding trajectory $\tau$
5:      Find set of new tasks to relabel with: $Z := \mathbb{S}(\tau)$
6:      Store original transitions in replay buffer: $(s_t, a_t, r(s_t, a_t, z), s_{t+1}, z)$
7:      **for** $z' \in Z$ **do**
8:          Store relabeled transitions in replay buffer: $(s_t, a_t, r(s_t, a_t, z'), s_{t+1}, z')$
9:      **end for**
10:     Perform $n$ steps of policy optimization with $\mathbb{A}$
11: **end for**

---

The core idea of hindsight relabeling is to convert the data generated from the policy under one task $z$ to a different task. Given the relabeled task $z' = \texttt{relabel}(\tau(\pi(.|z)))$, where $\tau$ represents the

trajectory induced by the policy $\pi(.|z)$, the state transition tuple $(s_t, a_t, r_t(.|z), s_{t+1})$ is converted to the relabeled tuple $(s_t, a_t, r_t(.|z'), s_{t+1})$. This relabeled tuple is then added to the replay buffer of an off-policy RL algorithm and trained as if the data generated from $z$ was generated from $z'$. If relabeling is done efficiently, it will allow for data that is sub-optimal under one reward specification $z$, to be used for the better relabeled specification $z'$. In the context of sparse goal-conditioned RL, where $z$ corresponds to a goal $g$ that needs to be achieved, HER [2] relabels the goal to states seen in the trajectory, i.e. $g' \sim \tau(\pi(.|z = g))$. This labeling strategy, however, only works in sparse goal conditioned tasks. In the following section, we describe two relabeling strategies that allow for a general application of hindsight.

## 3.2 Approximate IRL Relabeling (AIR)

The goal of computing the optimal reward parameter, given a trajectory is closely tied to the Inverse Reinforcement Learning (IRL) setting. In IRL, given demonstrations from an expert, we can retrieve the reward function the expert was optimized for. At the heart of these IRL algorithms, a reward specification parameter $z'$ is optimized such that

$$R(\tau_E|z') \geq R(\tau'|z') \; \forall \tau' \qquad (2)$$

where $\tau_E$ is an expert trajectory. Inspired by the IRL framework, we propose the *Approximate IRL* relabeling seen in Algorithm 2. We can use a buffer of past trajectories to find the task $z'$ on which our current trajectory does better than the older

---

**Algorithm 2** $\mathbb{S}_{IRL}$:    Approximate IRL

1: **Input:** Trajectory $\tau = (s_0, a_0, ..., s_T)$, cached reference trajectories $\mathcal{D} = \{(s_0, a_0, ..., s_T)\}_{i=1}^N$, reward function $r : \mathcal{S} \times \mathcal{A} \times \mathcal{T} \to \mathbb{R}$, number of candidate task variables to try: $K$, number of task variables to return: $m$
2: Sample set of candidate tasks $Z = \{v_j \sim \mathcal{T}\}_{j=1}^K$
    **Approximate IRL Strategy:**
3: **for** $v_j \in Z$ **do**
4:     **Calculate trajectory reward** for $\tau$ and the trajectories in $\mathcal{D}$: $R(\tau|v_j) := \sum_{t=0}^T \gamma^t r(s_t, a_t, v_j)$
5:     **Calculate percentile estimate:**
      $\hat{P}(\tau, v_j) = \frac{1}{n} \sum_{i=1}^N \mathbb{1}\{R(\tau|v_j) \geq R(\tau_i|v_j)\}$
6: **end for**
7: **return** $m$ tasks $v_j$ with highest percentiles $\hat{P}(\tau, v_j)$

---

ones. Intuitively this can be seen as an approximation of the right hand side of Eq. 2. Concretely, we want to relabel a new trajectory $\tau$, and have $N$ previously sampled trajectories along with $K$ randomly sampled candidate tasks $v_k$. Then, the relabeled task for trajectory $\tau$ is computed as:

$$z' = \arg\max_k \frac{1}{N} \sum_{j=1}^N \mathbb{1}\{R(\tau|v_k) \geq R(\tau_j|v_k)\} \qquad (3)$$

The relabeled $z'$ for $\tau$ maximizes its percentile among the $N$ most recent trajectories collected with our policy. One can also see this as an approximation of max-margin IRL [58]. One potential challenge with large $K$ is that many $v_k$ will have the same percentile. To choose between these potential task relabelings, we add tiebreaking based on the advantage estimate

$$\hat{A}(\tau, z) = R(\tau|z) - V^\pi(s_0, z) \qquad (4)$$

Among candidate tasks $v_k$ with the same percentile, we take the tasks that have higher advantage estimate. From here on, we will refer to Generalized Hindsight with Approximate IRL Relabeling as AIR.

## 3.3 Advantage Relabeling

One potential problem with AIR is that it requires $O(KNT)$ time to compute the relabeled task variable for each new trajectory, where $K$ is the number of candidate tasks, $N$ is the number of past trajectories compared to, and $T$ is the horizon. A relaxed version of AIR could significantly reduce computation time, while maintaining relatively high-accuracy relabeling. One way to do this is to use the *Maximum-Reward* relabeling objective. Instead of choosing from

---

**Algorithm 3** $\mathbb{S}_A$:    Trajectory Advantage

1: Repeat steps 1 & 2 from Algorithm 2
    **Advantage Relabeling Strategy:**
2: **for** $v_j \in Z$ **do**
3:     **Calculate trajectory advantage estimate**:
      $\hat{A}(\tau, v_j) = R(\tau|v_j) - V^\pi(s_0, v_j)$
4: **end for**
5: **return** $m$ tasks $z_j$ with highest $\hat{A}(\tau, z_j)$

---

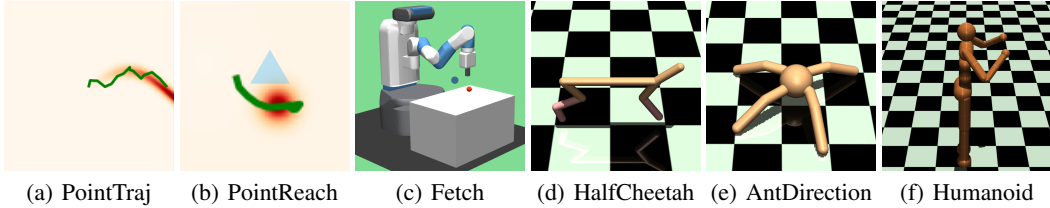

| (a) PointTraj | (b) PointReach | (c) Fetch | (d) HalfCheetah | (e) AntDirection | (f) Humanoid |

Figure 2: Environments we report comparisons on. PointTrajectory requires a 2D pointmass to follow a target trajectory; PointReacher requires moving the pointmass to a goal location, while avoiding an obstacle and modulating its energy usage. In (b), the red circle indicates the goal location, while the blue triangle indicates an imagined obstacle to avoid. Fetch has the same reward formulation as PointReacher, but requires controlling the noisy Fetch robot in 3 dimensions. HalfCheetah requires learning running in both directions, flipping, jumping, and moving efficiently. Ant and Humanoid require moving in a target direction as fast as possible.

our $K$ candidate tasks $v_k \sim \mathcal{T}$ by selecting for high percentile (Equation 2), we could relabel based on the cumulative trajectory reward:

$$z' = \arg\max_{v_k}\{R(\tau|v_k)\} \tag{5}$$

However, one challenge with simply taking the *Maximum-Reward* relabel is that different reward parameterizations may have different scales which will bias the relabels to a specific $z$. Say for instance there exists a task in the reward family $v_j$ such that $r(.|v_j) = 1 + \max_{i \neq j} r(.|v_i)$. Then, $v_j$ will always be the relabeled reward parameter irrespective of the trajectory $\tau$. Hence, we should not only care about the $v_k$ that maximizes reward, but select $v_k$ such that $\tau$'s likelihood under the trajectory distribution drawn from the optimal $\pi^*(.|v_k)$ is high. To do this, we can simply select $z'$ based on the advantage term that we used to tiebreak for AIR.

$$z'_i = \arg\max_k R(\tau|v_k) - V^\pi(s_0, v_k) \tag{6}$$

We call this *Advantage* relabeling (Algorithm 3), a more efficient, albeit less accurate, version of AIR. Empirically, *Advantage* relabeling often performs as well as AIR and has a runtime of only $O(KT)$, but relies on the value function $V^\pi$ more than AIR does. We reuse the twin $Q$-networks from SAC as our value function estimator.

$$V^\pi(s, z) = \min(Q_1(s, \pi(s|z), z), Q_2(s, \pi(s|z), z)) \tag{7}$$

In our experiments, we simply select $m = 1$ task out of $K = 100$ sampled task variables for all environments and both relabeling strategies. .

## 4 Experimental Evaluation

In this section, we describe our environments and discuss our central hypothesis: does relabeling improve performance? We also compare generalized hindsight against HER and a concurrently released hindsight relabeling algorithm, and examine the accuracy of different relabeling strategies.

### 4.1 Environments

Multi-task RL with a generalized family of reward parameterizations does not have existing benchmark environments. However, since sparse goal-conditioned RL has benchmark environments [55], we build on their robotic manipulation framework to make our environments. The key difference in the environment setting between ours and Plappert et al. [55] is that in addition to goal reaching, we have a dense reward parameterization for practical aspects of manipulation like energy consumption [42] and safety [10]. We show our environments in Figure 2 and clarify their dynamics and rewards in Appendix B. These environments will be released for open-source access.

### 4.2 Does Relabeling Help?

To understand the effects of relabeling, we compare our technique with the following standard baseline methods:

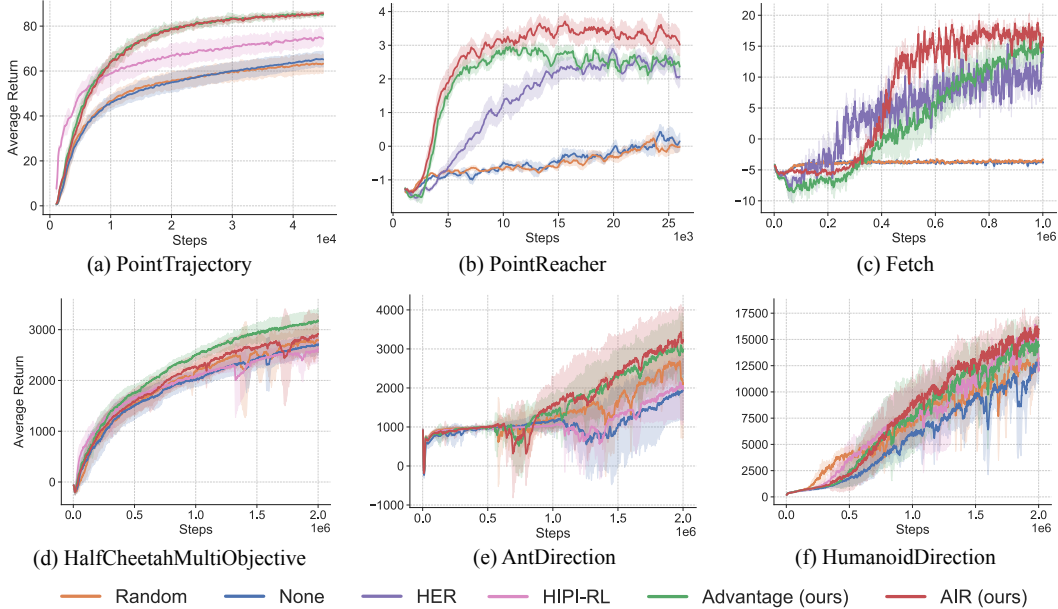

Figure 3: Learning curves comparing Generalized Hindsight algorithms to baseline methods. For environments with a goal-reaching component, we also compare to HER. The error bars show the standard deviation of the performance across 10 random seeds.

- No relabeling (None): as done in Yu et al. [71], we train with SAC without any relabeling.

- Intentional-Unintentional Agent (Random) [9]: when there is only a finite number of tasks, IU relabels a trajectory with every task variable. Since our space of tasks is continuous, we relabel with random $z' \sim \mathcal{T}$. This allows for information to be shared across tasks, albeit in a more diluted form. We perform further analysis of this baseline in Section 4.4.

- HER: for goal-conditioned tasks, we use HER to relabel the goal portion of the task with the future relabeling strategy. We leave the non-goal portion unchanged.

- HIPI-RL [17]: a concurrently released method for multi-task relabeling that resamples $z'$ every batch using a relabeling distribution proportional to the exponentiated $Q$-value. We discuss the differences between GH and HIPI-RL in Section 4.5.

We compare the learning performance for AIR and Advantage Relabeling with these baselines on our suite of environments in Figure 3. On all tasks, AIR and Advantage Relabeling outperform the baselines in both sample-efficiency and asymptotic performance. Both of our relabeling strategies outperform the Intentional-Unintentional Agent, implying that selectively relabeling trajectories with a few carefully chosen $z'$ is more effective than relabeling with many random tasks. Collectively, these results show that AIR can greatly improve learning performance, even on highly dense environments such as HalfCheetah, where learning signal is readily available. Advantage performs at least as well as AIR on all environments except PointReacher, Fetch, and Humanoid, where its performance is close. Thus, Advantage may be preferable in many scenarios, as it is 5-15% faster to train.

### 4.3 How does Generalized Hindsight compare to HER?

HER is, by design, limited to goal-reaching environments. For environments such as HalfCheetahMultiObjective, HER cannot be applied to relabel the weights on velocity, rotation, height, and energy. However, we can compare AIR with HER on the partially goal-reaching environments PointReacher and Fetch. Figure 3 shows that AIR achieves higher asymptotic performance than HER on both these environments. Figure 4 demonstrates on PointReacher how AIR can better choose the non-goal-conditioned parts of the task. Both HER and AIR place the relabeled goal around the terminus of the trajectory. However, only AIR understands that the imagined obstacle should be placed above the goal, since this trajectory becomes an optimal example of how to reach the new goal while avoiding the obstacle. HER has no such mechanism for precisely choosing an interesting

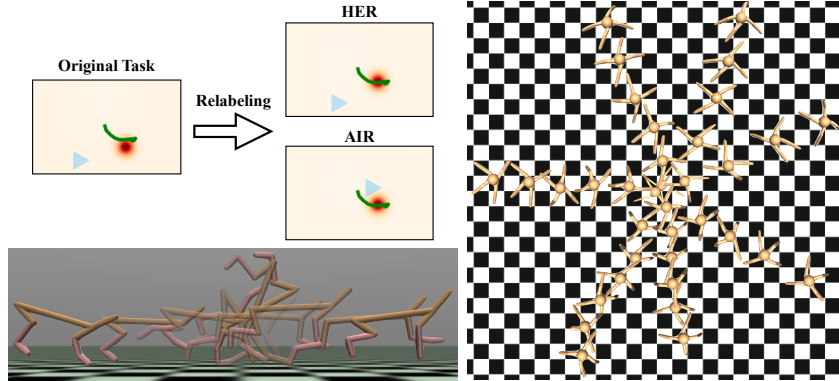

Figure 4: Top left: comparison of AIR vs HER. Red denotes the goal, and blue the obstacle. AIR places the relabeled obstacle within the curve of the trajectory, since this is the only way that the curved path would be better than a straight-line path (that would come close to the relabeled obstacle). Right and bottom left: visualizations of learned behavior on Ant and Half Cheetah, respectively.

obstacle location, since the obstacle does not affect the agent's ability to reach the goal. Thus, HER either leaves the obstacle in place or randomly places it, and it learns more slowly as a result.

## 4.4 Hindsight Bias and Random Relabeling

Off-policy reinforcement learning assumes that the transitions $(s_t, a_t, s_{t+1})$ are drawn from the distribution defined by the current policy and the transition probability function. Hindsight relabeling changes the distribution of transitions in our replay buffer, introducing *hindsight bias* for stochastic environments. This bias has been documented to harm sample-efficiency for HER [37], and is likely detrimental towards the performance of Generalized Hindsight.

In this section, we examine the tradeoffs between seeing more relevant data with relabeling, and incurring hindsight bias. A particularly interesting baseline is to randomly replace 80% of each minibatch with random tasks. In principle, this occasionally relabels transitions with good tasks, while avoiding hindsight bias. Note that this is different from the IU baseline, which relabels each transition only once.

We show results in Figure 5. Continual random relabeling accelerated learning in the first 25% of training relative to the IU baseline in the paper, but saturated to roughly the same asymptotic performance. There may be several reasons why this baseline doesn't match GH's performance.

First, random relabeling makes it difficult to see data that can be used to improve the policy. Later in training, random relabeling rarely provides any novel transitions that are better than those from the policy. This explains why learning plateaus and it underperforms GH: transitions are matched with the right tasks an increasingly tiny fraction of the time.

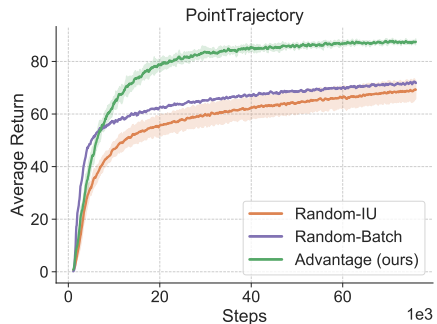

Figure 5: Comparing random relabeling strategies.

Second, the distribution of data in each minibatch is far from the state-action distribution of our policy. Fujimoto et al. [21] study a similar mismatch problem when training off-policy methods using data collected from a different policy (here, relabeled data is from the policy for a different task). Random relabeling introduces large training mismatch error, whereas any Bellman backup error from approximate IRL relabeling is optimistic and will encourage exploration in those areas of the MDP. In goal-reaching tasks, HER introduces small training mismatch error, since the relabeled transitions are always relevant to the relabeled goal. Similarly, as long as we balance "true" transitions and transitions relabeled with GH, we can obtain significant boosts in learning, even if we introduce hindsight bias. This need for balance is why we add each trajectory into the replay buffer with the original task $z$, in addition to a relabeled $z'$, and is likely why previous relabeling methods Andrychowicz et al. [2] and Nair et al. [47] relabel only 80% and 50% of the transitions, respectively. In more stochastic

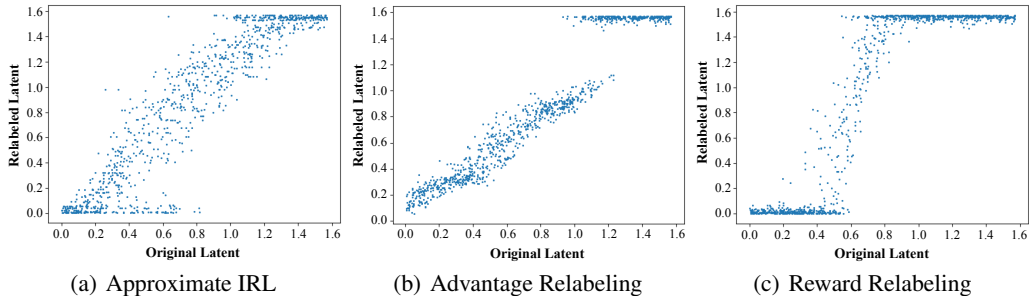

|(a) Approximate IRL|(b) Advantage Relabeling|(c) Reward Relabeling|

Figure 6: Comparison of relabeling fidelity on optimal trajectories. We roll out a trained PointReacher policy on 1000 randomly sampled tasks $z$, and apply each relabeling method to select from $K = 100$ randomly sampled tasks $v$. For approximate IRL, we compare against $N = 10$ prior trajectories. The x-axis shows the weight on energy for the task $z$ used for the rollout, while the y-axis shows the weight on energy for the relabeled task $z'$. Note that goal location, obstacle location, and weights on their rewards/penalties are varying as well, but are not shown. Closer to to the line $y = x$ indicates higher fidelity, since it implies $z' \approx z^*$.

environments, we would likely need to address hindsight bias by extending methods like ARCHER [37], or by applying the opposite of our method to present failed trajectories for each task, in addition to successful ones, as negative samples. Future work on gaining a better theoretical understanding of hindsight bias and the tradeoffs of relabeling would be highly useful for designing better algorithms.

## 4.5 Comparison to HIPI-RL

Eysenbach et al. [17], concurrently and independently, have released a method with similar motivation based on max-entropy IRL [72], rather than max-margin IRL [58] as ours is. Their method (HIPI-RL) repeatedly relabels each transition by sampling $z$ from the optimal relabeling distribution of tasks $q(z|s_t, a_t)$ that minimizes $D_{KL}(q(z, \tau)||p(z, \tau))$, where $p$ is the target joint distribution over tasks and trajectories.

$$q(z|s_t, a_t) \propto e^{Q(s,a,z) - \log Z(z)} \tag{8}$$

where $Z(z)$ is the per-task normalizing constant.

We compare HIPI-RL to GH in Figure 3, and find that ours is more sample-efficient and may be preferable for several reasons. Using the $Q$-function to relabel in HIPI-RL presents a chicken-and-the-egg problem where the $Q$-function needs to know that $Q(s, a, z)$ is large before we relabel the transition with $z$; however, it is difficult for the $Q$-function to do so unless the policy is already good and nearby data is plentiful. Furthermore, relabeling each transition is slow — if we see each transition an average of 10 times, then this uses 10 times the relabeling compute. Empirically, HIPI-RL takes roughly $4\times$ longer to train overall for simpler environments like PointTrajectory. HIPI-RL can even take up to $11\times$ longer on complex environments as HalfCheetah, Ant, and Humanoid, for about 2 weeks of training time. These environments require large batch sizes to stabilize training, which means that we need to do more transition relabeling overall. In comparison, Generalized Hindsight can be interpreted as doing maximum-likelihood estimation of the task $z$ on a trajectory-level basis. This is computationally faster and may have higher relabeling fidelity, due to lower reliance on the accuracy of the $Q$-function.

## 4.6 Analysis of Relabeling Fidelity

Approximate IRL, advantage relabeling, and reward relabeling are all approximate methods for finding the optimal task $z^*$ that a trajectory is (close to) optimal for. As a result, an important characteristic is their *fidelity*, i.e. how close the $z'$ they choose is to the true $z^*$. In Figure 6, we compare the fidelities of these three algorithms. Approximate IRL comes fairly close to reproducing the true $z^*$, albeit a bit noisily because it relies on the comparison to $N$ past trajectories. In the limit, as $N$ approaches infinity, AIR would find $z' = z^*$ perfectly, since it directly maximizes the max-margin IRL objective when comparing against infinite random trajectories in the limit. Thus, the cache size $N$ for AIR should be chosen to balance the relabeling computation time against the relabeling fidelity.

Advantage relabeling is slightly more precise, but fails for large energy weights, likely because the value function is not precise enough to differentiate between these tasks. Finally, reward relabeling does poorly, since it naively assigns $z'$ solely based on the trajectory reward, not how close the trajectory reward is to being optimal.

## 5 Related Work

### 5.1 Multi-task, Transfer, and Hierarchical Learning

Learning models that can share information across tasks has been concretely studied in the context multi-task learning [11], where models for multiple tasks are simultaneously learned. More recently, Kokkinos [35] and Doersch and Zisserman [14] look at shared learning across visual tasks, while Devin et al. [13] and Pinto and Gupta [53] look at shared learning across robotic tasks. Transfer learning [50, 68] focuses on transferring knowledge from one domain to another. One of the simplest forms of transfer is finetuning [22], where instead of learning a task from scratch it is initialized on a different task. Several other works look at more complex forms of transfer [70, 28, 4, 60, 36, 18, 23, 29]. In the context of RL, transfer learning [67] research has focused on learning transferable features across tasks [51, 5, 49]. One line of work by [59, 31, 13] has focused on network architectures that improves transfer of RL policies. Hierarchical reinforcement learning [44, 6] is another framework amenable for multi-task learning. Here the key idea is to have a hierarchy of controllers. One such setup is the Options framework [66] where the higher level controllers break down a task into sub-tasks and choose a low-level controller to complete each sub-task. Unsupervised learning of general low-level controllers has been a focus of recent research [19, 16, 62]. Variants of the Options framework [20, 39] can learn transferable primitives that can be used across a wide variety of tasks, either directly or after finetuning. Hierarchical RL can also be used to quickly learn how to perform a task across multiple agent morphologies [26]. All of these techniques are complementary to our method. They can provide generalizability to different dynamics and observation spaces, while Generalized Hindsight can provide generalizability to different reward functions.

### 5.2 Hindsight in Reinforcement Learning

Hindsight methods have been used for improving learning across as variety of applications. Andrychowicz et al. [2] use hindsight to efficiently learn on sparse, goal-conditioned tasks [54, 52, 3]. Nair et al. [47] approach goal-reaching with visual input by using hindsight relabeling within a learned latent space encoding for images. Several hierarchical methods [38, 45] train a low-level policy to achieve subgoals and a higher-level controller to propose those subgoals. These methods use hindsight relabeling to help the higher-level learn, even when the low-level policy fails to achieve the desired subgoals. Generalized Hindsight could be used to allow for richer low-level reward functions, potentially allowing for more expressive hierarchical policies.

### 5.3 Inverse Reinforcement Learning

Inverse reinforcement learning (IRL) has had a rich history of solving challenging robotics problems [1, 48]. More recently, powerful function approximators have enabled more general purpose IRL. For instance, Ho and Ermon [27] use an adversarial framework to approximate the reward function. Li et al. [40] extend this idea by learning reward functions on demonstrations from a mixture of experts. Our relabeling strategies currently build on top of max-margin IRL [58], but our central idea is orthogonal to the choice of IRL technique. Indeed, as discussed in subsection 4.5, Eysenbach et al. [17] concurrently and independently apply max-entropy IRL [72] towards relabeling. Future work should examine what scenarios each approach is best suited for.

## 6 Conclusion

In this work, we have presented Generalized Hindsight, a relabeling algorithm for multi-task RL based on approximate IRL. We demonstrate how efficient relabeling strategies can significantly improve performance on simulated navigation and manipulation tasks. Through these first steps, we believe that this technique can be extended to multi-task learning in other domains like real world robotics, where a balance between different specifications, such as energy use or safety, is important.

## Broader Impact

Our work investigates how to perform sample-efficient multi-task reinforcement learning. Generally, this goes against the trend of larger models and compute-hungry algorithms, such as state-of-the-art results in Computer Vision [12], NLP [8], and RL [69].

This will have several benefits in the short term. Better sample efficiency decreases the training time required for researchers to run experiments and for engineers to train models for production. This reduces the carbon footprint of the training process, and increases the speed at which scientists can iterate and improve on their ideas. Our algorithm enables autonomous agents to learn to perform a wide variety of tasks at once, which widens the range of feasible applications of reinforcement learning. Being able to adjust the energy consumption, safety priority, or other reward hyperparameters will allow these agents to adapt to changing human preferences. For example, autonomous cars may be able to learn to how avoid obstacles and adjust their driving style based on passenger needs.

Although our work helps make progress towards generalist RL systems, reinforcement learning remains impractical for most real-world problems. Reinforcement learning capabilities may drastically increase in the future, however, with murkier impacts. RL agents operating in the real world could improve the world by automating elderly care, disaster relief, cleaning and disinfecting, manufacturing, and agriculture. These agents could free people from menial, physically taxing, or dangerous occupations. However, as with most technological advances, developments in reinforcement learning could exacerbate income inequality, far more than the industrial or digital revolutions have, as profits from automation go to a select few. Reinforcement learning agents are also susceptible to reward misspecification, optimizing for an outcome that we do not truly want. Police robots instructed to protect the public may achieve this end by enacting discriminary and oppressive policies, or doling out inhumane punishments. Autonomous agents also increase the technological capacity for warfare, both physical and digital. Escalating offensive capabilities and ceding control to potentially uninterpretable algorithms raises the risk for international conflict to end in human extinction. Further work in AI alignment, interpretability, and safety is necessary to ensure that the benefits of strong reinforcement learning systems outweigh their risks.

## Acknowledgments and Disclosure of Funding

We thank AWS for computing resources. We also gratefully acknowledge the support from Berkeley DeepDrive, NSF, and the ONR Pecase award.

## Footnotes

[1]Website: sites.google.com/view/generalized-hindsight

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
