[Supplementary Material 1 · paper+appendix.pdf]

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

# A Preliminaries

A more comprehensive introduction to RL can be found in Kaelbling et al. [30] and Sutton and Barto [65].

## A.1 Reinforcement Learning

In this paper we deal with continuous space Markov Decision Processes $\mathcal{M}$ that can be represented as the tuple $\mathcal{M} \equiv (\mathcal{S}, \mathcal{A}, \mathcal{P}, r, \gamma, \mathbb{S})$, where $\mathcal{S}$ is a set of continuous states and $\mathcal{A}$ is a set of continuous actions, $\mathcal{P} : \mathcal{S} \times \mathcal{A} \times \mathcal{S} \to \mathbb{R}$ is the transition probability function, $r : \mathcal{S} \times \mathcal{A} \to \mathbb{R}$ is the reward function, $\gamma$ is the discount factor, and $\mathbb{S}$ is the initial state distribution.

An episode for the agent begins with sampling $s_0$ from the initial state distribution $\mathbb{S}$. At every timestep $t$, the agent takes an action $a_t = \pi(s_t)$ according to a policy $\pi : \mathcal{S} \to \mathcal{A}$. At every timestep $t$, the agent gets a reward $r_t = r(s_t, a_t)$, and the state transitions to $s_{t+1}$, which is sampled according to probabilities $\mathcal{P}(s_{t+1}|s_t, a_t)$. The goal of the agent is to maximize the expected return $\mathbb{E}_\mathbb{S}[R_0|\mathbb{S}]$, where the return is the discounted sum of the future rewards $R_t = \sum_{i=t}^{\infty} \gamma^{i-t} r_i$. The $Q$-function is defined as $Q^\pi(s_t, a_t) = E[R_t|s_t, a_t]$.

## A.2 Off Policy RL using Soft Actor Critic

Generalized Hindsight requires an off-policy RL algorithm to perform relabeling. One popular off-policy algorithm for learning deterministic continuous action policies is Deep Deterministic Policy Gradients (DDPG) [41]. The algorithm maintains two neural networks: the policy (also called the actor) $\pi_\theta : \mathcal{S} \to \mathcal{A}$ (with neural network parameters $\theta$) and a $Q$-function approximator (also called the critic) $Q_\phi^\pi : \mathcal{S} \times \mathcal{A} \to \mathbb{R}$ (with neural network parameters $\phi$).

During training, episodes are generated using a noisy version of the policy (called behaviour policy), e.g. $\pi_b(s) = \pi(s) + \mathcal{N}(0, 1)$, where $\mathcal{N}$ is the Normal distribution noise. The transition tuples $(s_t, a_t, r_t, s_{t+1})$ encountered during training are stored in a replay buffer [43]. Training examples sampled from the replay buffer are used to optimize the critic. By minimizing the Bellman error loss $\mathcal{L}_c = (Q(s_t, a_t) - y_t)^2$, where $y_t = r_t + \gamma Q(s_{t+1}, \pi(s_{t+1}))$, the critic is optimized to approximate the $Q$-function. The actor is optimized by minimizing the loss $\mathcal{L}_a = -\mathbb{E}_s[Q(s, \pi(s))]$. The gradient of $\mathcal{L}_a$ with respect to the actor parameters is called the deterministic policy gradient [63] and can be computed by backpropagating through the combined critic and actor networks. To stabilize the training, the targets for the actor and the critic $y_t$ are computed on separate versions of the actor and critic networks, which change at a slower rate than the main networks. A common practice is to use a Polyak averaged [56] version of the main network. Soft Actor Critic (SAC) [24] builds on DDPG by adding an entropy maximization term in the reward. Since this encourages exploration and empirically performs better than most actor-critic algorithms, we use SAC for our experiments, although Generalized Hindsight is compatible with any off-policy RL algorithm.

# B Environment Descriptions

**PointTrajectory**:

- Dynamics: This environment requires controlling a pointmass on a 2D plane, where the state $s_t = (x_t, y_t)$ represents the location of the pointmass. It always starts at the origin, with $s_0 = (0, 0)$, and the agent is restricted to the box $[-1, 1]^2$ via clipping. The agent can take actions $(dx, dy) \in [-0.1, 0.1]^2$, which affect the state via $s_{t+1} = (x_t + dx, y_t + dy)$.

- Rewards: The agent is rewarded for following a sinusoidal trajectory parameterized by a three-dimensional vector $z = (\theta, d, a)$. $\theta$ controls the orientation of the sinusoidal trajectory, $d$ controls its wavelength, and $a$ controls its amplitude. Specifically, the reward function is:

$$r(s, a|z) = \begin{cases} \frac{\tilde{x}}{d}\phi(\frac{\tilde{y} - a \times sin(\pi\tilde{x}/d)}{0.05}) & \text{if } \tilde{x} \geq 0 \\ 0 & \text{if } \tilde{x} < 0 \end{cases}$$

where $\tilde{x}$ is the projection of the current state $(x, y)$ onto the line $y = \tan(\theta)x$, $\tilde{y}$ is the orthogonal component, and $\phi$ is the probability density function of the unit Gaussian. The

$x/d$ term encourages movement towards the goal, and the $\phi(\cdot)$ term sharply penalizes an agent for deviating from the target trajectory.

The task distribution $\mathcal{T}$ is as follows: each element of $z$ is drawn independently from a separate distribution, where $\theta \sim \text{Unif}[-\pi, \pi]$, $d \sim \text{Unif}[0.75, 1]$, and $a \sim \text{Unif}[-0.25, 0.25]$.

**PointReacher**

- Dynamics: PointReacher shares the same dynamics as PointTrajectory.

- Rewards: This environment requires managing three quantities of interest: distance to a goal, distance from an obstacle, and energy used. We control these via a 6-dimensional task vector $z = (x_g, y_g, x_{obst}, y_{obst}, u, v)$, where the goal is located at $(x_g, y_g)$, the obstacle is located at $(x_{obst}, y_{obst})$, and $u$ and $v$ control the relative weighting of the three terms. Specifically, $u$ and $v$ represent a location on a unit sphere, and we calculate the weights $w_1$, $w_2$, and $w_3$ as the Euclidean coordinates of that point. The distance reward $r_{goal,t}$, safety reward $r_{obst,t}$, and energy penalty $E_t$ are calculated as following:

$$r_{goal,t} = 2 \exp \left\{ \frac{-((x_t - x_g)^2 + (y_t - y_g)^2)}{0.08^2} \right\}$$
$$r_{obst,t} = \log_{10} \left( 0.01 + (x_t - x_{obst})^2 + (y_t - y_{obst})^2 \right)$$
$$E_t = -||a_t||_2$$

We found that using these formulations encouraged specificity for the goal (i.e. optimal behavior is to move to the goal, not just a relatively close location), and a heavy penalty for coming too close to the obstacle.

Overall, the reward is then:

$$r(s_t, a_t | z) = w_1 r_{goal,t} + w_2 E_t + w_3 r_{obst,t}$$

Note that the obstacle is not physically present in the environment, so the dynamics of the environment are the same across all possible tasks. The obstacle is solely a part of the reward function, and the agent is discouraged from coming nearby via the penalty in the reward function. In general, this represents the idea that we can "practice" for certain safety-critical applications without needing to actually interact with the dangerous obstacle at hand.

The task distribution $\mathcal{T}$ is as follows: $u, v$ are sampled uniformly from the portion of the sphere in the first octant, and $(x_g, y_g)$ and $(x_{obst}, y_{obst})$ are sampled uniformly from the disk of radius 0.3 centered at the start state $s_0 = (0, 0)$.

**Fetch**

- Dynamics: We use the Fetch Robot from OpenAI Gym [7, 55]. Fetch has 3-dimensional observations, corresponding to the coordinates of its end-effector. It is operated through noisy position control, so the action space is also 3-dimensional. We increase the number of solver iterations from 20 to 100, to make the controller more accurate and reduce control noise.

- Rewards: We use the same parameterized reward function as we do in PointReacher, adapted so that the goal and obstacle lie in 3D space.

**HalfCheetahMultiObjective**

- Dynamics: We use the HalfCheetah-v1 environment from OpenAI Gym [7], corresponding to a two-legged robot constrained to run in the $xz$ plane. It has a 17-dimensional observation space, and a 6-dimensional action space of torque inputs for each joint.

- Rewards: Our task variable $z \in \mathcal{Z} \subseteq \mathbb{R}^4$ controls a weighted combination of velocity in the x-direction ($v_t$), energy use ($E_t$), height of the center of mass ($h_t$), and rotation speed ($\omega_t$).

$$r(s_t, a_t | z) = z_1 v_t + z_2 E_t + z_3 h_t + z_4 \omega_t$$

where $v_t = x_{t+1} - x_t$, $\omega_t = \theta_{t+1} - \theta_t$, and $E_t = -0.1 \times ||a_t||_2^2$. Since only the ratio between individual elements of $z$ are relevant for eliciting different behavior, we constrain

the set of allowed task variables $\mathcal{Z} = \{z \in \mathbb{R}^4 : ||z||_2 = 1, z_1 \geq 0, z_2 \geq 0\}$. We enforce non-negativity of $z_1$ and $z_2$ because it is rather unimpressive behavior to maximize energy use without purpose or minimize the height of the robot's center of mass. To sample from the task distribution $\mathcal{T}$, we sample uniformly from $\mathcal{Z}$.

**AntDirection**

- Dynamics: We use the Ant-v2 environment from OpenAI Gym [7], corresponding to a four-legged robot that can move in the $x$, $y$, and $z$ directions. The state space is 111-dimensional, and the joints are controlled via torque control, for an 8-dimensional action space.

- Rewards: We use a 1-dimensional task variable to control which direction the Ant robot should move in. Our reward function is:

$$r(s, a|z) = ||\text{velocity}||_2 \times \mathbb{1}\{\text{velocity angle within 15 degrees of z}\}$$

This reward function encourages moving quickly in the direction chosen by the task variable $z$. An alternative reward function that returns the length of the velocity component in the direction of $z$ allows a policy to gain high rewards while going even at a $45°$ angle from the target direction, so we add the indicator function to restrict the space of high-reward behavior. The task distribution $\mathcal{T}$ is uniform over $\mathcal{Z} = [-\pi, \pi]$.

**HumanoidDirection**

- Dynamics: We use the Humanoid-v2 environment from OpenAI Gym [7], corresponding to a bipedal humanoid robot that can move and rotate in all directions. The state space is 376-dimensional, and the joints are controlled via torque control, for a 17-dimensional action space.

- The task family here is similar to the one for AntDirection, with small tweaks. In addition to the standard survival bonus and the torque and contact force penalty, we add a scaled form of the velocity reward from AntDirection.

$$r_{vel}(s, a|z) = 10 \times ||\text{velocity}||_2 \times \mathbb{1}\{\text{velocity angle within 15 degrees of z}\}$$

Empirically, the $10\times$ factor on the velocity is necessary for learning a proper task-conditional at all. Without it, the learned policy stands still for every learning algorithm, from no relabeling to AIR.

## C  Training Details

### C.1  Training Tricks

Since our task space $\mathcal{Z}$ is continuous, we cannot use multi-headed networks, which are commonly used when dealing with a discrete set of tasks [71]. We thus follow the ubiquitous approach of concatenating $(s||z)$ and feeding that into the policy network $\pi$, and concatenating $(s||a||z)$ and feeding that into the q-network. However, when the observation space is large, such as in Ant (111-dimensional) or HalfCheetah (17-dimensional), it becomes difficult for the network to identify the few dimensions of its input that correspond to $z$ and should thus strongly determine the desired behavior (for $\pi$) or the correct $Q$-value.

We deal with this by (a) repeating the task variable $z$ a few times, and (b) appending the repeated latent at every hidden layer. (a) increases the salience of $z$, which should improve the ability of the network to do credit assignment and identify $z$ as the causal variable responsible for the difference between tasks. (b) allows the policy and q networks to more easily have differentiated behavior, depending on the current task $z$. Empirically, we find that these two tricks are crucial for getting the baselines (No relabeling, Intentional-Unintentional Agent) to work at all; our Generalized Hindsight methods benefit to a smaller degree. For HalfCheetahMultiObjective, AntDirection, and HumanoidDirection, we repeat the task variable 5 times.

### C.2  Hyperparameters

We list shared hyperparameters in Table 1, and environment-specific hyperparameters in Table 2.

| Parameter | Value |
|---|---|
| Algorithm | Soft Actor Critic [24] |
| Optimizer | Adam [34] |
| Batch size | 256 |
| Target smoothing coefficient ($\tau$) | 0.005 |
| Reward scale | Auto-tuned [25] |

Table 1: Hyperparameters used for the experiments shown in Figure 3.

| Environment | Hidden Layers | Learning Rate | Updates/Epoch | Horizon | Discount ($\gamma$) | AIR Cache ($N$) |
|---|---|---|---|---|---|---|
| PointTrajectory | [400, 300] | $3 \times 10^{-3}$ | 100 | 15 | 0.9 | 500 |
| PointReacher | [400, 300] | $3 \times 10^{-3}$ | 200 | 20 | 0.97 | 500 |
| Fetch | [400, 300] | $3 \times 10^{-3}$ | 200 | 50 | 0.98 | 500 |
| HalfCheetah | [256, 256] | $3 \times 10^{-4}$ | 1000 | 1000 | 0.99 | 50 |
| AntDirection | [256, 256] | $3 \times 10^{-4}$ | 1000 | 1000 | 0.99 | 50 |
| HumanoidDirection | [256, 256] | $3 \times 10^{-4}$ | 1000 | 1000 | 0.99 | 50 |

Table 2: Environment-specific hyperparameters used for the experiments shown in Figure 3.

# D   Videos

A video of our environments and results can be found here: sites.google.com/view/generalized-hindsight.

[Supplementary Material 2]

# A Preliminaries

A more comprehensive introduction to RL can be found in Kaelbling et al. [30] and Sutton and Barto [65].

## A.1 Reinforcement Learning

In this paper we deal with continuous space Markov Decision Processes $\mathcal{M}$ that can be represented as the tuple $\mathcal{M} \equiv (\mathcal{S}, \mathcal{A}, \mathcal{P}, r, \gamma, \mathbb{S})$, where $\mathcal{S}$ is a set of continuous states and $\mathcal{A}$ is a set of continuous actions, $\mathcal{P} : \mathcal{S} \times \mathcal{A} \times \mathcal{S} \to \mathbb{R}$ is the transition probability function, $r : \mathcal{S} \times \mathcal{A} \to \mathbb{R}$ is the reward function, $\gamma$ is the discount factor, and $\mathbb{S}$ is the initial state distribution.

An episode for the agent begins with sampling $s_0$ from the initial state distribution $\mathbb{S}$. At every timestep $t$, the agent takes an action $a_t = \pi(s_t)$ according to a policy $\pi : \mathcal{S} \to \mathcal{A}$. At every timestep $t$, the agent gets a reward $r_t = r(s_t, a_t)$, and the state transitions to $s_{t+1}$, which is sampled according to probabilities $\mathcal{P}(s_{t+1}|s_t, a_t)$. The goal of the agent is to maximize the expected return $\mathbb{E}_{\mathbb{S}}[R_0|\mathbb{S}]$, where the return is the discounted sum of the future rewards $R_t = \sum_{i=t}^{\infty} \gamma^{i-t} r_i$. The $Q$-function is defined as $Q^\pi(s_t, a_t) = E[R_t|s_t, a_t]$.

## A.2 Off Policy RL using Soft Actor Critic

Generalized Hindsight requires an off-policy RL algorithm to perform relabeling. One popular off-policy algorithm for learning deterministic continuous action policies is Deep Deterministic Policy Gradients (DDPG) [41]. The algorithm maintains two neural networks: the policy (also called the actor) $\pi_\theta : \mathcal{S} \to \mathcal{A}$ (with neural network parameters $\theta$) and a $Q$-function approximator (also called the critic) $Q_\phi^\pi : \mathcal{S} \times \mathcal{A} \to \mathbb{R}$ (with neural network parameters $\phi$).

During training, episodes are generated using a noisy version of the policy (called behaviour policy), e.g. $\pi_b(s) = \pi(s) + \mathcal{N}(0, 1)$, where $\mathcal{N}$ is the Normal distribution noise. The transition tuples $(s_t, a_t, r_t, s_{t+1})$ encountered during training are stored in a replay buffer [43]. Training examples sampled from the replay buffer are used to optimize the critic. By minimizing the Bellman error loss $\mathcal{L}_c = (Q(s_t, a_t) - y_t)^2$, where $y_t = r_t + \gamma Q(s_{t+1}, \pi(s_{t+1}))$, the critic is optimized to approximate the $Q$-function. The actor is optimized by minimizing the loss $\mathcal{L}_a = -\mathbb{E}_s[Q(s, \pi(s))]$. The gradient of $\mathcal{L}_a$ with respect to the actor parameters is called the deterministic policy gradient [63] and can be computed by backpropagating through the combined critic and actor networks. To stabilize the training, the targets for the actor and the critic $y_t$ are computed on separate versions of the actor and critic networks, which change at a slower rate than the main networks. A common practice is to use a Polyak averaged [56] version of the main network. Soft Actor Critic (SAC) [24] builds on DDPG by adding an entropy maximization term in the reward. Since this encourages exploration and empirically performs better than most actor-critic algorithms, we use SAC for our experiments, although Generalized Hindsight is compatible with any off-policy RL algorithm.

# B Environment Descriptions

**PointTrajectory**:

- Dynamics: This environment requires controlling a pointmass on a 2D plane, where the state $s_t = (x_t, y_t)$ represents the location of the pointmass. It always starts at the origin, with $s_0 = (0, 0)$, and the agent is restricted to the box $[-1, 1]^2$ via clipping. The agent can take actions $(dx, dy) \in [-0.1, 0.1]^2$, which affect the state via $s_{t+1} = (x_t + dx, y_t + dy)$.
- Rewards: The agent is rewarded for following a sinusoidal trajectory parameterized by a three-dimensional vector $z = (\theta, d, a)$. $\theta$ controls the orientation of the sinusoidal trajectory, $d$ controls its wavelength, and $a$ controls its amplitude. Specifically, the reward function is:

$$r(s, a|z) = \begin{cases} \frac{\tilde{x}}{d} \phi\left(\frac{\tilde{y} - a \times sin(\pi \tilde{x}/d)}{0.05}\right) & \text{if } \tilde{x} \geq 0 \\ 0 & \text{if } \tilde{x} < 0 \end{cases}$$

where $\tilde{x}$ is the projection of the current state $(x, y)$ onto the line $y = \tan(\theta)x$, $\tilde{y}$ is the orthogonal component, and $\phi$ is the probability density function of the unit Gaussian. The

$x/d$ term encourages movement towards the goal, and the $\phi(\cdot)$ term sharply penalizes an agent for deviating from the target trajectory.

The task distribution $\mathcal{T}$ is as follows: each element of $z$ is drawn independently from a separate distribution, where $\theta \sim \text{Unif}[-\pi, \pi]$, $d \sim \text{Unif}[0.75, 1]$, and $a \sim \text{Unif}[-0.25, 0.25]$.

## PointReacher

- Dynamics: PointReacher shares the same dynamics as PointTrajectory.

- Rewards: This environment requires managing three quantities of interest: distance to a goal, distance from an obstacle, and energy used. We control these via a 6-dimensional task vector $z = (x_g, y_g, x_{obst}, y_{obst}, u, v)$, where the goal is located at $(x_g, y_g)$, the obstacle is located at $(x_{obst}, y_{obst})$, and $u$ and $v$ control the relative weighting of the three terms. Specifically, $u$ and $v$ represent a location on a unit sphere, and we calculate the weights $w_1$, $w_2$, and $w_3$ as the Euclidean coordinates of that point. The distance reward $r_{goal,t}$, safety reward $r_{obst,t}$, and energy penalty $E_t$ are calculated as following:

$$r_{goal,t} = 2 \exp \left\{ \frac{-((x_t - x_g)^2 + (y_t - y_g)^2)}{0.08^2} \right\}$$
$$r_{obst,t} = \log_{10} \left( 0.01 + (x_t - x_{obst})^2 + (y_t - y_{obst})^2 \right)$$
$$E_t = -||a_t||_2$$

We found that using these formulations encouraged specificity for the goal (i.e. optimal behavior is to move to the goal, not just a relatively close location), and a heavy penalty for coming too close to the obstacle.

Overall, the reward is then:

$$r(s_t, a_t | z) = w_1 r_{goal,t} + w_2 E_t + w_3 r_{obst,t}$$

Note that the obstacle is not physically present in the environment, so the dynamics of the environment are the same across all possible tasks. The obstacle is solely a part of the reward function, and the agent is discouraged from coming nearby via the penalty in the reward function. In general, this represents the idea that we can "practice" for certain safety-critical applications without needing to actually interact with the dangerous obstacle at hand.

The task distribution $\mathcal{T}$ is as follows: $u, v$ are sampled uniformly from the portion of the sphere in the first octant, and $(x_g, y_g)$ and $(x_{obst}, y_{obst})$ are sampled uniformly from the disk of radius 0.3 centered at the start state $s_0 = (0, 0)$.

## Fetch

- Dynamics: We use the Fetch Robot from OpenAI Gym [7, 55]. Fetch has 3-dimensional observations, corresponding to the coordinates of its end-effector. It is operated through noisy position control, so the action space is also 3-dimensional. We increase the number of solver iterations from 20 to 100, to make the controller more accurate and reduce control noise.

- Rewards: We use the same parameterized reward function as we do in PointReacher, adapted so that the goal and obstacle lie in 3D space.

## HalfCheetahMultiObjective

- Dynamics: We use the HalfCheetah-v1 environment from OpenAI Gym [7], corresponding to a two-legged robot constrained to run in the $xz$ plane. It has a 17-dimensional observation space, and a 6-dimensional action space of torque inputs for each joint.

- Rewards: Our task variable $z \in \mathcal{Z} \subseteq \mathbb{R}^4$ controls a weighted combination of velocity in the x-direction ($v_t$), energy use ($E_t$), height of the center of mass ($h_t$), and rotation speed ($\omega_t$).

$$r(s_t, a_t | z) = z_1 v_t + z_2 E_t + z_3 h_t + z_4 \omega_t$$

where $v_t = x_{t+1} - x_t$, $\omega_t = \theta_{t+1} - \theta_t$, and $E_t = -0.1 \times ||a_t||_2^2$. Since only the ratio between individual elements of $z$ are relevant for eliciting different behavior, we constrain

the set of allowed task variables $\mathcal{Z} = \{z \in \mathbb{R}^4 : ||z||_2 = 1, z_1 \geq 0, z_2 \geq 0\}$. We enforce non-negativity of $z_1$ and $z_2$ because it is rather unimpressive behavior to maximize energy use without purpose or minimize the height of the robot's center of mass. To sample from the task distribution $\mathcal{T}$, we sample uniformly from $\mathcal{Z}$.

**AntDirection**

- Dynamics: We use the Ant-v2 environment from OpenAI Gym [7], corresponding to a four-legged robot that can move in the $x$, $y$, and $z$ directions. The state space is 111-dimensional, and the joints are controlled via torque control, for an 8-dimensional action space.

- Rewards: We use a 1-dimensional task variable to control which direction the Ant robot should move in. Our reward function is:

$$r(s, a|z) = ||\text{velocity}||_2 \times \mathbb{1}\{\text{velocity angle within 15 degrees of z}\}$$

This reward function encourages moving quickly in the direction chosen by the task variable $z$. An alternative reward function that returns the length of the velocity component in the direction of $z$ allows a policy to gain high rewards while going even at a $45°$ angle from the target direction, so we add the indicator function to restrict the space of high-reward behavior. The task distribution $\mathcal{T}$ is uniform over $\mathcal{Z} = [-\pi, \pi]$.

**HumanoidDirection**

- Dynamics: We use the Humanoid-v2 environment from OpenAI Gym [7], corresponding to a bipedal humanoid robot that can move and rotate in all directions. The state space is 376-dimensional, and the joints are controlled via torque control, for a 17-dimensional action space.

- The task family here is similar to the one for AntDirection, with small tweaks. In addition to the standard survival bonus and the torque and contact force penalty, we add a scaled form of the velocity reward from AntDirection.

$$r_{vel}(s, a|z) = 10 \times ||\text{velocity}||_2 \times \mathbb{1}\{\text{velocity angle within 15 degrees of z}\}$$

Empirically, the $10\times$ factor on the velocity is necessary for learning a proper task-conditional at all. Without it, the learned policy stands still for every learning algorithm, from no relabeling to AIR.

## C  Training Details

### C.1  Training Tricks

Since our task space $\mathcal{Z}$ is continuous, we cannot use multi-headed networks, which are commonly used when dealing with a discrete set of tasks [71]. We thus follow the ubiquitous approach of concatenating $(s||z)$ and feeding that into the policy network $\pi$, and concatenating $(s||a||z)$ and feeding that into the q-network. However, when the observation space is large, such as in Ant (111-dimensional) or HalfCheetah (17-dimensional), it becomes difficult for the network to identify the few dimensions of its input that correspond to $z$ and should thus strongly determine the desired behavior (for $\pi$) or the correct $Q$-value.

We deal with this by (a) repeating the task variable $z$ a few times, and (b) appending the repeated latent at every hidden layer. (a) increases the salience of $z$, which should improve the ability of the network to do credit assignment and identify $z$ as the causal variable responsible for the difference between tasks. (b) allows the policy and q networks to more easily have differentiated behavior, depending on the current task $z$. Empirically, we find that these two tricks are crucial for getting the baselines (No relabeling, Intentional-Unintentional Agent) to work at all; our Generalized Hindsight methods benefit to a smaller degree. For HalfCheetahMultiObjective, AntDirection, and HumanoidDirection, we repeat the task variable 5 times.

### C.2  Hyperparameters

We list shared hyperparameters in Table 1, and environment-specific hyperparameters in Table 2.

| Parameter | Value |
|---|---|
| Algorithm | Soft Actor Critic [24] |
| Optimizer | Adam [34] |
| Batch size | 256 |
| Target smoothing coefficient ($\tau$) | 0.005 |
| Reward scale | Auto-tuned [25] |

Table 1: Hyperparameters used for the experiments shown in Figure 3.

| Environment | Hidden Layers | Learning Rate | Updates/Epoch | Horizon | Discount ($\gamma$) | AIR Cache ($N$) |
|---|---|---|---|---|---|---|
| PointTrajectory | [400, 300] | $3 \times 10^{-3}$ | 100 | 15 | 0.9 | 500 |
| PointReacher | [400, 300] | $3 \times 10^{-3}$ | 200 | 20 | 0.97 | 500 |
| Fetch | [400, 300] | $3 \times 10^{-3}$ | 200 | 50 | 0.98 | 500 |
| HalfCheetah | [256, 256] | $3 \times 10^{-4}$ | 1000 | 1000 | 0.99 | 50 |
| AntDirection | [256, 256] | $3 \times 10^{-4}$ | 1000 | 1000 | 0.99 | 50 |
| HumanoidDirection | [256, 256] | $3 \times 10^{-4}$ | 1000 | 1000 | 0.99 | 50 |

Table 2: Environment-specific hyperparameters used for the experiments shown in Figure 3.

# D   Videos

A video of our environments and results can be found here: sites.google.com/view/generalized-hindsight.