[Reviews · NeurIPS 2020]

Review 1

Summary and Contributions: This paper introduces a new experience relabeling method for multi-task reinforcement learning. The authors' show how their approach leads to more and better sample re-use across multiple tasks in a multi-task setting, which leads to more efficient learning. ---------- after rebuttal------------ I'm happy with the way authors' addressed my comments and stand by my positive score.

Strengths: - The authors address a very important problem of multi-task reinforcement learning and sample re-usability - The authors do a good job analyzing the results and their algorithm. In particular, I enjoyed Sections 4.3 and 4.4 as well as the fidelity results and analysis in Fig. 5 - The authors provide a quantitative comparison to a very similar concurrent method and provide justification for the difference between the two.

Weaknesses: - The main weakness of the paper in my opinion is the lack of theoretical rigor to justify some of the claims as well as the language that is often imprecise. For example: -> The description of the method in line 55-56 is misleading in that it indicates that the original trajectory with the originally intended task is not used and it is relabeled instead. Later in the paper, in Section 3 and in the algorithm box, the authors explain that they use the original task as well as the relabeled one. - The proposed relabeling strategy only considers successful trajectories. In the extreme case, we could imagine a situation where there is a set of successful trajectories for one task (that was potentially collected with another task in mind). In this case, the authors' algorithm would always pick the successful trajectories even though we know that informative negatives are crucial for off-policy RL algorithms. There is no discussion of this case and of negatives in general in this work. - It is not clear why relabeling is done on the entire trajectory rather than on the individual transition. The authors only mention this at the very end of their paper when comparing to HIPI, please justify it earlier. - Fig. 3 e) is missing the learning curve for AIR - Comparison to the concurrent HIPI work is only done on a single task - it would be useful to add HIPI as an extra baseline for all the experiments.

Correctness: The authors claims are mostly correct with a few exceptions where there is not enough evidence or formal proves for their statements. In particular: - It is not clear why the Advantage Relabeling algorithm is less accurate than AIR. I think this is a speculation. There are no bounds that show how well both algorithms approximate the IRL problem and it's very unclear which one is more accurate in that regard. - The authors' statement about the lack of multi-task RL benchmarks in line 186 is not true. In fact, the authors cite one of them (Meta-World) [71]

Clarity: The paper is easy to read but it could be improved in terms of being more precise and provide justification to its claims as outlined above. Small error: In line 155 the authors introduce the AIR acronym, even though they started referring to this algorithm as AIR much earlier in the manuscript (line 53)

Relation to Prior Work: Yes, the authors also go the extra mile to explain the difference to a method developed concurrently (HIPI).

Reproducibility: Yes

Additional Feedback: I would encourage the authors to provide a larger discussion of the role of negatives in reinforcement learning and experience relabeling. In addition, I think that adding HIPI as a baseline to all the experiments would strengthen the paper. I also encourage the authors to improve the justification of some of their claims (e.g. why is AIR a better approximation to the IRL problem).


Review 2

Summary and Contributions: The authors present an extension of Hindsight Experience Replay (HER), which makes it possible to use continuous task parameters. They motivate their work with multi-task RL and see their contribution as one way to reduce the sample complexity there. As they state in the introduction, the main contributions are a) to extend HER to “reward families”; b) to propose a relabelling algorithm that they call AIR; c) to demonstrate significant improvements in multi-task RL on a suite of multi-task navigation and manipulation tasks. On a meta level it assumes a single agent to operate on a family of MDPs M(z), that are parameterised by a vector z, which configures the reward function r(z) for one instance of the MDP. For each collected episode they try to find a set of “good” matching z and add the same data relabelled with a new reward. The authors propose two methods how a set of appropriate z can be found: AIR, which is motivated by inverse reinforcement learning, and a method based on the advantage function. While AIR performs better on most experiments, it is computationally more expensive. The authors compare both methods on a set of experiments against a baseline without relabelling and an Intentional-Unintentional agent with random goal parameters. Code to reproduce the results is available in the supplementary material.

Strengths: The presented method is a nice and practical extension to HER. It is nicely motivated and builds logically on the ideas and achievements of the basic method. All statements are clear and technical sound. Both methods to sample are clearly described and the experiments are technical sound, with rich supplementary material. The main claim of the paper can be stated as: Relabelling helps to learn more data efficiently and also helps to reach a better final performance. This main claim is supported by the experiments.

Weaknesses: The most critical point is probably similar methods that were published recently. The authors discuss one of them in section 4.5, but they only briefly touch it and compare it only in a single experiment. Here the reader would certainly like this as a baseline method in all experiments (main evaluation section). In my opinion the paper could be improved by adding these comparisons. It makes a nice idea and a nice paper. But it is very close to the ideas of HER and the presented ideas for sampling new reward parameters are straight forward. A better discussion of various parameters of the sampling process could help to improve here. All in all, the novelty aspect for a conference like NeurIPS seems borderline in the current state.

Correctness: All results, the description and derivation of the method, as well as the empirical methodology look correct and suitable. Some smaller improvements, see general comments.

Clarity: Most of the paper are written in a clear way, well structured and good to understand. Especially the method is developed and explained so that the reader knows what the idea is and how it works. Section 4.3 to 4.5 were harder to read / understand and would require a bit of work to make them more structured and to sharpen the statements .

Relation to Prior Work: The authors embed their work in the most important literature and other approaches in the field. As mentioned above, the relation to a recently published paper by Eisenbach et al. (2020) is included as section 4.5, but the reader probably wants to see this method as a baseline for all of the experiments.

Reproducibility: Yes

Additional Feedback: Some smaller things to improve: in Algorithm 1 and Algorithm 2 you describe how to sample. But you only describe it as “sample a set of z”, I didn’t catch how many to sample. I would assume that this is an important parameter that you have to define, maybe also ablate (also for the baselines). The method - as it is presented - appears to be tailored to goal conditioned settings. It uses the “reward family” that would in principle allow more diverse tasks, but all experiments are mainly reacher tasks with different goals or walking tasks with different target directions. It would be also nice for the reader if you would discuss relations to multi-task approaches with a more diverse set of tasks. (e.g. Riedmiller et al.: Learning by Playing Solving Sparse Reward Tasks from Scratch, IXML 2018) What I really wondered as a reader: Would the “reward family” generalise to more diverse tasks like “grasp”, “reach”? In principle the tasks are very close to goal conditioned tasks, where we “just” modify the target value. Sec 1: “Recent work from” … 2017, maybe wrong citation? Sec 4.4, Figure 4: It would need more explanation for the reader to understand this. Especially why HER places the obstacle right while AIR doesn’t? Figure 3, e: comparison to HER missing?


Review 3

Summary and Contributions: This paper proposes a new hindsight data utilization scheme for helping multi-task RL. Different from previous hindsight for relabeling the learning goals, this paper proposes to relabel reward functions with different tasks for the generated trajectories. To achieve this, two algorithms, based on IRL, are developed to identify the suited tasks for the trajectories. Experiments demonstrate the proposed algorithm performs better than baselines. ====== after rebuttal: I'm happy with my original positive score.

Strengths: - The paper is generally written clearly and easy to follow. - The proposed generalized hindsight scheme is interesting. - Two algorithms for relabeling the trajectories are developed and the second one somehow addresses the issues of high computation cost. - Experiments demonstrate the proposed algorithm works better than baselines with significant advantage.

Weaknesses: - Though the paper is motivated for solving multi-task RL, the experiment setting is not for multi-task RL. Whether the proposed algorithm can perform well for real multi-task RL problems, as illustrated in the method part, is not very clear. - There are some hyperparameters (e.g. number of tasks returned m) that need to be specified when deploying AIR for task re-labeling. It is unclear how mis-specified hyperparameters affect the performance. E.g. some tasks that are not very suited for the trajectories are chosen for labeling the trajectories may bias and hurt the quality of the learned task policy. - The trajectories may be generated from a different policy parameter from that one being optimized, How to solve such underlying distribution drift problem?

Correctness: Yes

Clarity: Yes

Relation to Prior Work: Yes

Reproducibility: Yes

Additional Feedback:


Review 4

Summary and Contributions: The authors describe a generalization of hindsight relabeling for training in RL to enable applying the technique to problems with non-sparse rewards. The paper then describes two specific algorithms that accomplish this with different approximation vs compute tradeoffs and empirically compares them to Hindsight Experience Replay (HER), a naive random goal selection baseline, and a no-hindsight baseline.

Strengths: * Paper is well written and organized. * The proposed method was evaluated on simulated environments that are plausibly relevant to the RL community and generally outperformed all baselines.

Weaknesses: * Experiments are only conducted with six non-standard multi-task problems (though they are derived from standard ones). See detailed comment under ‘Correctness’.

Correctness: The claims and methodology look sound overall; some minor potential issues are listed under 'Additional feedback'.

Clarity: Paper is largely well organized and easy to understand.

Relation to Prior Work: The authors clearly contrast their methods with existing work -- most notably Hindsight Experience Replay (HER) and a recent preprint by Eisenbach et al. The rephrasing of the hindsight approach appears to be novel as do the two specific proposed methods.

Reproducibility: Yes

Additional Feedback: * The learning curves used to compare the proposed methods to baselines show error bars derived from ten sample runs, but I didn’t notice anything specifying what this error bar signifies. Relatedly, given that some of these error bars have non-trivial overlap, ten trials may not be sufficient. * In order to highlight the improved generality of their proposed method, the authors mostly use customized environments with dense rewards and goal-conditioned tasks where the task is represented by more than the goal state (e.g. alternate obstacle arrangements). Some readers might reasonably expect to see the proposed methods evaluated on some purely goal-reaching multi-task environments as well as some environments with pure sparse rewards. * Authors might consider making clearer which evaluation environments include dense rewards and which are partially goal-reaching (and therefore somewhat amenable to HER) vs which meta-tasks are purely characterized by non-state parameters.

[Author Response · NeurIPS 2020]

We thank the reviewers for their feedback. We are excited to see that all four reviewers have rated our work positively
and found that our paper addresses a very important problem (R1), is nicely motivated and builds logically on prior
work (R2), works significantly better than baselines (R3), and is clearly contrasted with existing work (R4). We address
the most salient points of feedback below, and will incorporate all of their feedback in the final version of our paper.

──────────────────────── **General Feedback** ────────────────────────

**Novelty** (R1, R2). "R2: *similar methods ... were published recently. It is very close to ... HER and the presented ideas*
*for sampling new reward parameters are straightforward.*" Indeed, the power of our work is that we present simple
methods to extend HER to general multi-task problems. Work by Eysenbach et al. 2020 was concurrent with ours, and
is likely simultaneously in submission to NeurIPS 2020. Nevertheless, we contrast our work with HIPI as follows: our
work explores different environments, has a different connection to inverse RL, and has extensive comparison of the
pros and cons of each method. We're running HIPI as a baseline for all of our environments, which we will add when
complete. However, Fig. 6 shows HIPI is worse on easy environments while taking more time to train; hence it is hard
to see it working for more complex settings. Computationally, HIPI is $\approx 11\times$ slower than our method on environments
that require bigger batch sizes. On Ant and Humanoid, this requires almost two weeks to run, which is limiting in
academic labs such as ours. This analysis of computational complexity will be added to the paper.

**Tasks** (R3, R4). "R4: *the authors mostly use customized environments with dense rewards and goal-conditioned tasks*
*where the task is represented by more than the goal state.*" For sparse goal-reaching, HER should work better, since
the optimal relabeled task can be easily computed from the states reached. Our algorithms are designed for settings
beyond pure goal reaching, where HER cannot be straightforwardly applied. We will clarify the categories of rewards
that go into each environment, beyond the explicit reward formulae in Appendix B. And although our methods may not
accelerate learning two highly dissimilar tasks, e.g. hammering a nail and opening a door, we show that they work well
on a variety of meaningful and challenging robotics problems, such as control with safety or energy trade-offs.

**Theory** (R1). "R1: *There are no bounds that show how well both algorithms approximate the IRL problem and it's very*
*unclear which one is more accurate.*" As with most prior work in hindsight relabeling [2, 46, 38, 45], we found it difficult
to prove any meaningful bounds, likely due to the complexity of handling a distribution of policies $\{\pi(\cdot|z)|z \in \mathcal{Z}\}$ and
the accuracy of the $Q$-function. Both are complicated by function approximation and complex dynamics and rewards.
Intuitively, AIR may be more accurate because AIR directly maximizes the max-margin IRL objective when comparing
against infinite random trajectories in the limit. We compared our relabeling algorithms empirically in Sec 4.4, and
would love to see future work advancing our theoretical understanding of hindsight algorithms.

──────────────────────── **Algorithmic / Experimental details** ────────────────────────

**Negative trajectories** (R1). "R1: *the authors' algorithm would always pick the successful trajectories even though we*
*know that informative negatives are crucial for off-policy RL algorithms.*" Indeed, hindsight relabeling in stochastic
environments introduces hindsight bias, a tradeoff we discuss in Appendix C. In spite of this, our relabeling methods
achieve good performance in the robotics environments that we test in. Extending our methods to more stochastic
environments, e.g. by applying ARCHER [37], is an exciting avenue for future work.

**Relabeling trajectories vs transitions** (R1). "R1: *It is not clear why relabeling is done on the entire trajectory rather*
*than on the individual transition.*" It's possible to relabel single transitions, based on (a) the transition reward or (b) the
$Q$-value, but these both have problems. (a) Relabeling just based on the 1-step reward produces myopic relabeling that
values short term rewards over large rewards that accumulate later. (b) In Sec 4.5, we present an in-depth discussion of
the problems of using the $Q$-value to do transition relabeling. Relabeling the entire trajectory once is computationally
efficient, simple, and already provides large performance gains. We will add this discussion earlier in the paper.

**Distribution shift** (R3). "R3: *The trajectories may be generated from a different policy parameter from that one*
*being optimized.*" This is mostly fine for off-policy learning algorithms, such as SAC. We discuss bootstrap error in
Appendix C, where we note that distribution shift from relabeled trajectories should at worst result in overestimation
and encourage exploration of promising areas of the MDP. Empirically, our methods still perform well, and a promising
extension of our work is to use offline RL methods to further reduce distribution shift error.

**Hyperparameters for relabeling** (R4). "R4: *There are some hyperparameters (e.g. number of tasks returned m) that*
*need to be specified when deploying AIR.*" We simply select $m = 1$ relabeled trajectories out of $K = 100$ sampled task
variables for all environments, and will add an ablation study on the effect of $m$ and $K$ in the camera-ready's appendix.
With $m = 1$, our relabeling algorithms are empirically accurate in finding the proper relabeled task, as seen in Sec 4.4.

**Error bars** (R4). "R4: *I didn't notice anything specifying what this error bar signifies.*" The error bars show the
standard deviation across seeds. When corrected by $1/\sqrt{n}$ for the number of runs, they have minimal overlap.

We again thank the reviewers for their detailed reviews, and even pointing out a few typos and missing citations, which
we will add to the final version of the paper.

[Meta-Review · NeurIPS 2020]

Reviewers were unanimously positive about this manuscript's clarity and contribution, and while acknowledging its shortcomings, all felt there was at least a weak case for acceptance. R1 & R2 were positive about the author rebuttal and I'd encourage the authors to incorporate their addressing of reviewers' concerns into the camera ready.